# Metriplectic Structure of a Radiation–Matter-Interaction Toy Model

**DOI:** 10.3390/e24040506

**Published:** 2022-04-04

**Authors:** Massimo Materassi, Giulia Marcucci, Claudio Conti

**Affiliations:** 1Institute for Complex Systems, National Research Council (ISC-CNR), Via Madonna del Piano 10, 50019 Sesto Fiorentino, Italy; 2Institute for Space Astrophysics and Planetology (IAPS), National Institute of Astrophysics (INAF), 00133 Rome, Italy; 3Apoha Ltd., 242 Acklam Rd., London W10 5JJ, UK; irreversiblegm@gmail.com; 4Department of Physics, University Sapienza, Piazzale Aldo Moro 5, 00185 Rome, Italy; claudio.conti@cnr.it; 5Institute for Complex Systems, National Research Council (ISC-CNR), Via dei Taurini 19, 00185 Rome, Italy

**Keywords:** two-photon absorption, metriplectic systems, dissipative systems, asymptotically stable equilibrium, Madelung variables

## Abstract

A dynamical system defined by a metriplectic structure is a dissipative model characterized by a specific pair of tensors, which defines a Leibniz bracket; and a free energy, formed by a “Hamiltonian” and an entropy, playing the role of dynamics generator. Generally, these tensors are a Poisson bracket tensor, describing the Hamiltonian part of the dynamics, and a symmetric metric tensor, that models purely dissipative dynamics. In this paper, the metriplectic system describing a simplified two-photon absorption by a two-level atom is disclosed. The Hamiltonian component is sufficient to describe the free electromagnetic radiation. The metric component encodes the radiation–matter coupling, driving the system to an asymptotically stable state in which the excited level of the atom is populated due to absorption, and the radiation has disappeared. First, a description of the system is used, based on the real–imaginary decomposition of the electromagnetic field phasor; then, the whole metriplectic system is re-written in terms of the phase–amplitude pair, named Madelung variables. This work is intended as a first result to pave the way for applying the metriplectic formalism to many other irreversible processes in nonlinear optics.

## 1. Introduction

The modeling of irreversible systems is a fundamental issue in every physical field. Even though in quantum mechanics this is a still debated topic, and many efforts have been made both in cases of intrinsic irreversibility [1] and in open systems [2], classical mechanics boasts many more tools and much more established and recognized theories to describe time-asymmetric phenomena.

Even if the paramount majority of space physics, geophysical, ecological and biological processes are of dissipative nature, the strongest formalism in theoretical physics, namely the Hamiltonian formalism, is not able to describe dissipation. In Hamiltonian formalism, the structure of a physical system and its interactions are all encoded in the quantity *H* named “Hamiltonian”, so that many statements about the dynamics are possible by the simple inspection of this quantity. Moreover, the system physics is “algebrized”, and any transformation of the system description is implemented in the simple language of Poisson brackets, defined via a Jacobi tensor. All these powerful tools of analytical mechanics fail for dissipative systems. An extension of the Hamiltonian formalism may be defined for dissipative systems, the so-called *metriplectic formalism* [3]. The role of the Hamiltonian *H* is played by the free energy *F* of the system, while the Poisson bracket is generalized by the *metriplectic bracket*, defined via two tensors. Metriplectic structures represent dissipative systems with a simple theory based on linear algebraic tools, that have immediate thermodynamical translation, both in classical [4] and in quantum [5] systems. In particular, the energy *F* extends the Hamiltonian *H* combining it with an entropic quantity *S*; the two tensors forming the metriplectic bracket are a Jacobi tensor *J*, and a semimetric tensor *G* related to the Onsager coefficients [6]. Metriplectic formalism is perfectly analogous to what is referred to as *General Equation for Non-Equilibrium Reversible-Irreversible Coupling* (GENERIC), described in [6], in which the role of the tensors *J* and *G* is strongly stressed.

In the literature, there are several examples of irreversible dynamics represented as metriplectic systems, from very simple systems in Newton’s mechanics [7], to hydrodynamics [8] and magneto-hydrodynamics [9]; more delicate, but extremely interesting, are the cases of kinetic equations, the collisional terms of which may be written as a semi-metric term, or that of a free rotator driven to a stable rotation axis by a suitably designed servo-engine [3,10]. In all the cases mentioned, the non-Hamiltonian system is gifted of the transparency of motions generated by Leibniz algebrae [11], even if proper symplectic formalism is not applicable; moreover, the energy landscape becomes tractable as the free energy of the system is explicitly written.

In the present paper, the metriplectic formalism is used to describe a process in which electromagnetic radiation is absorbed by the atoms of a medium. As the radiation disappears, the medium electrons go to a higher energy level, so that the final state of the system is an excited state of matter with no radiation. This process may be regarded as the irreversible sink of an ordered form of energy, contained in the incident electromagnetic field, into the many atoms of the medium, represented with the collective variable of the excited population. This pictorial view of the radiation absorption as a dissipative process is physically supported by the fact that the terms describing absorption in the equations of electromagnetism are time-asymmetric terms, isomorphic to those ones describing friction in mechanics.

The process of *two-photon absorption* (TPA) by a two-level atom is here described through a *classical* dynamical system, in which the energy initially located in the radiation variables is *irreversibly* converted into the energy pertaining to the population of the excited level. The final state, in which no free radiation exists any more while the excited state is populated, is *the asymptotic equilibrium state* of the system. The existence of asymptotically stable equilibrium makes the TPA similar to a *dissipative process*, such as macroscopic friction, where an “ordered” form of energy is “consumed” in favour of the “internal energy” of some medium. This attitude describes the matter absorbing the electromagnetic wave energy as the environment of a system that would be Hamiltonian per se; the presence of the environment, with the matter–radiation interaction that “destroys” the radiation, breaks the Hamiltonian nature of the radiation dynamics. Such a scenario is described by the extension of the symplectic algebra of the Hamiltonian system to a *metriplectic algebra of brackets* [3], where the Hamiltonian component of the motion is still given by the original Poisson bracket, while a suitable *semi-defined metric bracket* generates the non-Hamiltonian component. An extension of the Hamiltonian, namely the *free energy* of the system, represents the metriplectic generator of the motion. The foregoing program interprets the dynamics of classical dissipative systems as flows generated by a new kind of *Leibniz algebrae* of brackets [11], namely the *metriplectic bracket*.

It is important to state that the metriplectic formalism does not necessarily describe any possible dissipative system, and a wide class of dissipative systems might not be cast into this form. Yet, as mentioned above and in the following Section 2, many important dissipative systems are indeed of metriplectic nature.

This paper is organized as follows.

In Section 2, we review the metriplectic formalism from a very general point of view. In Section 3, radiation is described by the real and imaginary parts of a complex phasor ψ, namely the *q* and *p*, while the atomic population is described by a real variable *n*. The ODEs describing the evolution of *q*, *p* and *n* in the presence of the dissipative interaction are presented. Then, the dissipationless, i.e., Hamiltonian, limit is recovered, in which the expression of the free radiation energy $H_{0}\left(q,p\right)$ works as a Hamiltonian and the population *n* does not evolve. In Section 4, the metriplectic algebra generating the non-Hamiltonian component of the dynamics is constructed. First of all, equations are composed to define the semi-metric tensor through which the metric bracket $\left(\cdot ,\cdot \right)$ is defined; then, a completion energy U(n) is constructed in order for $H\left(q,p,n\right)=H_{0}\left(q,p\right)+U\left(n\right)$ to be constant along the non-Hamiltonian motion of the full system (q,p,n). Finally, the framework is completed by defining the proper conditions on the entropy S(n) and writing down the expression of the free energy F(q,p,n)=H(q,p,n)+χS(n). Equilibrium points are determined as a consequence of this construction (in the sense that, choosing different expressions for S(n), i.e., for F(q,p,n), different equilibria $n_{\mathrm{eq}}$ are found). Section 5 is devoted to the translation of the metriplectic system from the variable set $\left(q,p,n\right)$ to the Madelung set $\left(\phi ,\rho ,n\right)$, with ϕ and ρ being the phase–amplitude variables for the electromagnetic field. In Section 6, we summarize the results of our analysis and indicate some future possible developments. Details on the computation of the metric tensor *G* are added in the Appendix A, while in Appendix B a point is clarified about the particular form the tensor *G* assumes in this specific problem.

## 2. General Metriplectic Formalism

Before describing how the metriplectic formalism is applied to the TPA, it is useful to sketch briefly the construction of a metriplectic system.

Typically, one starts from a Hamiltonian system described by a set of variables *X*, the dynamics of which are generated by some Hamiltonian $H_{0}\left(X\right)$ and some Poisson bracket $\left\{\cdot ,\cdot \right\}$ so that $\dot{X}=\left\{X,H_{0}\left(X\right)\right\}$. Then, some quantity *S* is defined, with the property of being *in involution* with *any* possible function of *X*, $\left\{S,A\right\}=0\;\forall \;A\left(X\right)$, i.e., to be *a Casimir of $\left\{\cdot ,\cdot \right\}$*. This quantity *S* may either depend on the original variables *X* only (as for the kinetic theories or for the rigid body mentioned before), or on some “environmental” variable *Y* too (as in the case of a particle motion with friction, or those of non-ideal hydrodynamics or magneto-hydrodynamics; this will be the case here too). The Casimir *S* becomes the generator of a new non-Hamiltonian component of the motion through the introduction of a new bracket, $\left(\cdot ,\cdot \right)$, with properties of symmetry and semi-definiteness [3]. The extended system, based on the old Hamiltonian one, now has new dynamics in which the variables *X* evolve according to
(1)$$\dot{X}=\left\{X,H\left(X,Y\right)\right\}+\chi \left(X,S\left(X,Y\right)\right),$$
while the motion of the environmental variables, if any, is typically influenced by *S* and the metric bracket only:(2)$$\dot{Y}=\chi \left(Y,S\left(X,Y\right)\right).$$

Metriplectic systems describe the evolution of dissipative dynamics to *asymptotically stable equilibria*, so in general, with Z=(X,Y), there will exist some $Z_{\mathrm{eq}}$ to which Z(t) converges for t→+∞. The state $Z_{\mathrm{eq}}$ is the equilibrium towards which the system “thermalizes”.

In Equation (Equation 1), the Hamiltonian H(X,Y) may be different from the original $H_{0}\left(X\right)$, as it may include a term depending on *Y* in order to close the system energetically, and may take into account of the irreversible consumption of $H_{0}\left(X\right)$ (dissipation); the difference $U=H-H_{0}$ is the *internal energy* of the environment. In Equations (Equation 1) and (Equation 2), the factor χ is a coefficient representing a coupling condition between *X* and *Y*, and characterizing the asymptotically stable equilibrium (this χ is related to the thermodynamic temperature, when dissipation is due to some thermal bath); the strength of the dissipative interaction, defining the non-dissipative (Hamiltonian) regime in some suitable limit of its, is some α included in the definition of $\left(\cdot ,\cdot \right)$, so that α→0 turns off dissipation. The mathematical expressions (Equation 1), (Equation 2) of $\dot{X}$ and $\dot{Y}$ depend on this α, so one may well say:(3)$$\lim_{\alpha \to 0}\dot{X}\left(\alpha \right)=\left\{X,H_{0}\left(X,Y\right)\right\},\;\;\lim_{\alpha \to 0}\dot{Y}\left(\alpha \right)=0.$$ From Equations (Equation 1) and (Equation 2) it appears that the limit for χ→0 also gives the ODEs in Equation (Equation 3); however, this does not switch off dissipation, but simply describes a condition in which it is uneffective as in the condition of zero absolute temperature, see Section 4 and Section 6.

As far as the bracket $\left(\cdot ,\cdot \right)$ and the Casimir *S* are concerned, the semi-definiteness of the first one, $\left(A,B\right)\leq 0\;\forall \;A,B$, and a suitable choice of the sign of χ, i.e., χ<0, implies that *S* will grow monotonically along the system motion $\dot{S}\geq 0$, until some asymptotically stable equilibrium $Z_{\mathrm{eq}}=\left(X_{\mathrm{eq}},Y_{\mathrm{eq}}\right)$ is reached, so that $\dot{S}\left(Z_{\mathrm{eq}}\right)=0$ [3]. In other words, the Casimir *S* turns out to be *a Lyapunov function around $Z_{\mathrm{eq}}$*, and it can be understood as a form of *entropy* [12] (of course, all the reasoning just presented is rephrased “without *Y*” for those metriplectic systems in which no “environment” needs to be defined, as in the case of kinetic theories).

In order to complete the picture, the property $\left(H,A\right)=0\;\forall \;A$ is requested for the metric bracket and the total Hamiltonian *H*, in order for dissipation not to “delete” the total energy, but just transform it.

Regarding metripletcic systems, one further thing must be highlighted. Given the expression of the Poisson bracket $\left(\cdot ,\cdot \right)$, many Casimir functions *S* may exist; choosing different forms of *S*, the dynamics in Equations (Equation 1) and (Equation 2) will converge to different specific equilibria $Z_{\mathrm{eq}}$. It must be underlined that this construction does not include “all” the dynamical systems referred to as “metriplectic” in literature: this is the construction of a *complete metriplectic system* (CMS), while incomplete metriplectic systems (IMS) may be defined too, with the two brackets but the Hamiltonian as the only generator. IMS can be suitable tools to describe energetically open systems [4].

The development presented here makes the TPA process tractable in a very transparent way as a CMS, and points towards the systematic algebrization of non-linear optics.

The system we introduce here in order to turn the TPA process into a CMS has three degrees of freedom: two real variables describe the electromagnetic radiation, one real variable describes the matter. Radiation is represented either via a complex phasor ψ, or a couple of real variables, that in turn can be either the couple (q,p) of the real and imaginary part of ψ, or the couple $\left(\phi ,\rho \right)$ of its phase and (square root) amplitude; the population of the excited level is given by some real positive variable *n*. As mentioned before, the electromagnetic variable ψ, or (q,p) or (ϕ,ρ), are conceived as “mechanical variables”, representing the “exact” state of the electromagnetic wave, while the excited population *n* is more a thermodynamic variable. During the *irreversible process*, the electromagnetic energy $H_{0}\left(\psi \right)$ is converted into some energy U(n) associated with n≠0. In our “metriplectization” scheme, one starts from the equations of motion of the state Z=(q,p,n) and observes that a suitable limit of them reduces the system to a Hamiltonian one. In this Hamiltonian limit, a Poisson bracket is defined, so that *q* and *p* are canonically conjugated $\left\{q,p\right\}=1$, while *n* remains apparently outside the play as $\left\{n,q\right\}=\left\{n,p\right\}=0$. As the population of the excited level is *in involution* with *q* and *p*, *any* function S(n) will be a Casimir for $\left\{\cdot ,\cdot \right\}$. The program then is to find a suitable function $H_{0}\left(q,p\right)$ that may play the role of Hamiltonian in the Hamiltonian limit. This represents the radiation energy, to be extended as $H\left(q,p,n\right)=H_{0}\left(q,p\right)+U\left(n\right)$ to include the energy pertaining to the filling of the excited state, namely the internal energy of the environment “atoms”. In order to complete the metriplectic framework, suitable forms for S(n) and for the metric bracket $\left(\cdot ,\cdot \right)$ must be constructed, and this is essentially what is achieved in the present work. As anticipated, field variables $\left(q,p\right)$ may be replaced by some $\left(\phi ,\rho \right)$ in which $\dot{\rho }=0$ in the Hamiltonian limit and ϕ evolves linearly with time. Then, the whole CMS can be re-expressed in the new variables $\left(\phi ,\rho ,n\right)$.

## 3. Two-Photon Absorption Toy Model

Nonlinear optical systems such as those mediated by two-photon absorptions are described by a generalized nonlinear Schödinger equation including effects such as linear dispersion, dispersion of nonlinearity, and higher order nonlinear processes [12]. In our toy model, we limit it to consider only nonlinear effects that are the core of our metriplectic analysis and that can be cast in the simplest formulations. Specifically, we consider an electromagnetic field described by a complex envelope *A* ($I={\left|A\right|}^{2}$ is the optical intensity) that undergoes a self-phase modulation due to the optical Kerr effect. We also include absorption as a multiphoton process, such that we have a two-level system that absorbs two photons (see Figure 1).

Writing the complex refractive index nonlinear optical perturbation as $\Delta n=n_{2}{\left|A\right|}^{2}+i\alpha_{P}n$ with $n_{2}$ the Kerr coefficient, and assuming the optical beam propagating in the *z* direction, with vacuum wavenumber $k_{0}=2\pi /\lambda$, with λ the wavelength, the dynamic equation for the field envelope is
(4)$$\frac{\partial A}{\partial z}=ik_{0}\frac{\Delta n}{n}A=\frac{2\pi }{\lambda }(in_{2}{\left|A\right|}^{2}-\alpha_{P}n)A.$$

Seemingly, the equation for the level inversion reads
(5)$$\frac{\partial n}{\partial z}=k_{P}{\left|A\right|}^{4}$$
and the multiphoton coefficient is $k_{P}$.

Equations (Equation 4) and (Equation 5) can be cast in dimensionless units by scaling the evolution coordinate *z* with $z_{0}$, and letting $A=A_{0}\psi$, with $A_{0}^{2}=\lambda /\left(2\pi n_{2}z_{0}\right)$, and defining the dimensionless absorption coefficient $\alpha =2\pi \alpha_{p}z_{0}/\lambda$ and the dimensionless multiphoton coefficient as $k=2k_{P}z_{0}A_{0}^{4}$.

Hence, we consider a very simplified toy model for the two-level atomic system [13,14]. The TPA, sketched in Figure 1, is expressed by the following differential equations:(6)$$\left\{\begin{array}{c} \dot{\psi }=ı{\left|\psi \right|}^{2}\psi -\alpha n\psi ,\\ \dot{n}=\frac{k}{2}{\left|\psi \right|}^{4}, \end{array}\right.$$
where ψ is the complex field amplitude, *n* the population of the second level, k>0 and α>0. This system is directly derived by the multi-photon absorption model [15,16,17], when neglecting several physical phenomena, e.g., the spontaneous emission. Moving to real-valued functions, we define
(7)$$\psi =\frac{q-ıp}{\sqrt{2}},$$
so that in terms of the variables *q* and *p*, the system (Equation 6) reads:(8)$$\left\{\begin{array}{c} \dot{q}=\frac{1}{2}p\left(q^{2}+p^{2}\right)-\alpha nq,\\ \dot{p}=-\frac{1}{2}q\left(q^{2}+p^{2}\right)-\alpha np,\\ \dot{n}=\frac{k}{8}{\left(q^{2}+p^{2}\right)}^{2}. \end{array}\right.$$

The phasor ψ may be also represented by *Madelung variables*
$\left(\phi ,\rho ,n\right)$, as: (9)$$\psi =\sqrt{\rho }e^{ı\phi }.$$ These canonically conjugated ϕ and ρ describe the system as reported in Section 5.

It is useful to observe that in the limit
(10)α→0,k→0 Equation (Equation 8) become a Hamiltonian system, so that the conditions (Equation 10) will be referred to as *non-dissipative limit* (NDL); under these conditions, the ODEs in *q*, *p* and *n* read:(11)$$\left\{\begin{array}{c} \dot{q}=\frac{1}{2}p\left(q^{2}+p^{2}\right),\\ \dot{p}=-\frac{1}{2}q\left(q^{2}+p^{2}\right),\\ \dot{n}=0. \end{array}\right.$$ As one defines the Hamiltonian
(12)$$H_{0}=\frac{1}{2}{\left(\frac{q^{2}+p^{2}}{2}\right)}^{2}.$$
and the Poisson bracket
(13)$$\left\{q,p\right\}=1,\;\left\{q,n\right\}=0,\;\left\{p,n\right\}=0,$$
any quantity $f\left(q,p,n\right)$ evolves according to:$$\dot{f}=\left\{f,H_{0}\right\}$$
along the motion (Equation 11). The quantity defined in Equation (Equation 12) turns out to be the energy that can be attributed to the free radiation, as it is not interacting with matter.

The dissipative nature of the dynamical system emerges as one sees that the following relationships hold
(14)$${\dot{H}}_{0}=-4\alpha nH_{0},\;\dot{n}=kH_{0}$$
along the motions (Equation 8). As α and *k* are positive constants, and as long as n≥0, this means that $\dot{H_{0}}\leq 0$ and $\dot{n}\geq 0$. All in all, system (Equation 14), that is equivalent to Equation (Equation 8), simply describes the consumption of $H_{0}$ in favour of the quantity *n*. System (Equation 8) has energy dynamics similar to *classical dissipation*, which points towards the formulation of it as a *complete metriplectic system* [12].

## 4. Metriplectic Formulation

In order to recognize a CMS equivalent to Equations (Equation 8), let us put those ODEs in the general form of a Hamiltonian system “perturbed” by dissipative terms, the most general form of which reads:(15)$$\left\{\begin{array}{c} \dot{q}=\left\{q,H\right\}+\psi_{q},\\ \dot{p}=\left\{p,H\right\}+\psi_{p},\\ \dot{n}=\left\{n,H\right\}+\psi_{n}, \end{array}\right.$$
with H(p,q,n) the *total Hamiltonian* and $\left\{f,g\right\}=J^{ij}\partial_{i}f\partial_{j}g$ the *Poisson brackets* (PB) with
(16)$$J^{ij}=\left(\begin{array}{ccc} 0 & 1 & 0\\ -1 & 0 & 0\\ 0 & 0 & 0 \end{array}\right)$$ (here we have used i,j=q,p,n). Seeing that {n,A}=0 for any observable A(q,p,n) is straightforward from Equation (Equation 16). This implies that any function C(n) is a Casimir; indeed, $\left\{C\left(n\right),H\right\}=C^{\prime }\left(n\right)\left\{n,H\right\}=0$, with $C^{\prime }=\frac{dC}{dn}$, so that one has: $$(10)\Rightarrow \dot{C}\left(n\right)=0.$$

In order to express Equation (Equation 15) as a metriplectic system [12,18], we define the *metric brackets* $\left(f,g\right)=G^{ij}\partial_{i}f\partial_{j}g$, constructing
(17)$$G^{ij}=\left(\begin{array}{ccc} G^{qq} & G^{qp} & G^{qn}\\ G^{qp} & G^{pp} & G^{pn}\\ G^{qn} & G^{pn} & G^{nn} \end{array}\right)$$
as a symmetric, positive semi-definite matrix.

Equation (Equation 15) will be put in the form of
(18)$$\left\{\begin{array}{c} \dot{q}=\left\{q,H\right\}+\chi \left(q,S\right),\\ \dot{p}=\left\{p,H\right\}+\chi \left(p,S\right),\\ \dot{n}=\chi \left(n,S\right), \end{array}\right.$$
where χ is a constant to be calculated once the desired $Z_{\mathrm{eq}}$ is defined. Indeed, once defined, the *metriplectic Leibniz brackets*
(19)<<f,g>>:={f,g}+(f,g),
the *entropy*
S(q,p,n) and the *free energy*
F=H+χS, if ∇S∈Ker(J) and ∇H∈Ker(G), namely,
(20)$$J^{ij}\partial_{j}S=G^{ij}\partial_{j}H=0,$$
then, one has
(21)$$\dot{g}=<<g,F>>=\left\{g,H\right\}+\chi \left(g,S\right).$$ Equation (Equation 20) implies that the CMS entropy must be a mere function of *n*.

### 4.1. The Metric Brackets Tensor

Thanks to Equation (Equation 20), the system in Equation (Equation 18) becomes
(22)$$\left\{\begin{array}{c} \dot{q}=\partial_{p}H+\chi G^{qn}S^{\prime }\left(n\right),\\ \dot{p}=-\partial_{q}H+\chi G^{pn}S^{\prime }\left(n\right),\\ \dot{n}=\chi G^{nn}S^{\prime }\left(n\right), \end{array}\right.$$
therefore, our overriding concern is to determine the tensor *G*. In order to obtain such a result, we follow a linear algebraic procedure. Calculations are illustrated item-by-item in the Appendix A. The final result is:$$G_{E}^{qq}=\frac{\partial_{q}H\partial_{n}H\left[b\partial_{q}H\partial_{n}H+2c\partial_{p}H\sqrt{{\left(\partial_{q}H\right)}^{2}+{\left(\partial_{p}H\right)}^{2}+{\left(\partial_{n}H\right)}^{2}}\right]}{\left[{\left(\partial_{q}H\right)}^{2}+{\left(\partial_{p}H\right)}^{2}\right]\left[{\left(\partial_{q}H\right)}^{2}+{\left(\partial_{p}H\right)}^{2}+{\left(\partial_{n}H\right)}^{2}\right]},$$
$$G_{E}^{qp}=\frac{\partial_{n}H\left[b\partial_{q}H\partial_{p}H\partial_{n}H+c\left[{\left(\partial_{p}H\right)}^{2}-{\left(\partial_{q}H\right)}^{2}\right]\sqrt{{\left(\partial_{q}H\right)}^{2}+{\left(\partial_{p}H\right)}^{2}+{\left(\partial_{n}H\right)}^{2}}\right]}{\left[{\left(\partial_{q}H\right)}^{2}+{\left(\partial_{p}H\right)}^{2}\right]\left[{\left(\partial_{q}H\right)}^{2}+{\left(\partial_{p}H\right)}^{2}+{\left(\partial_{n}H\right)}^{2}\right]},$$
$$G_{E}^{qn}=-\frac{b\partial_{q}H\partial_{n}H+c\partial_{p}H\sqrt{{\left(\partial_{q}H\right)}^{2}+{\left(\partial_{p}H\right)}^{2}+{\left(\partial_{n}H\right)}^{2}}}{{\left(\partial_{q}H\right)}^{2}+{\left(\partial_{p}H\right)}^{2}+{\left(\partial_{n}H\right)}^{2}},$$
$$G_{E}^{pp}=\frac{b{\left(\partial_{p}H\right)}^{2}{\left(\partial_{n}H\right)}^{2}-2c\partial_{q}H\partial_{p}H\partial_{n}H\sqrt{{\left(\partial_{q}H\right)}^{2}+{\left(\partial_{p}H\right)}^{2}+{\left(\partial_{n}H\right)}^{2}}}{\left[{\left(\partial_{q}H\right)}^{2}+{\left(\partial_{p}H\right)}^{2}\right]\left[{\left(\partial_{q}H\right)}^{2}+{\left(\partial_{p}H\right)}^{2}+{\left(\partial_{n}H\right)}^{2}\right]},$$
$$G_{E}^{pn}=\frac{-b\partial_{p}H\partial_{n}H+c\partial_{q}H\sqrt{{\left(\partial_{q}H\right)}^{2}+{\left(\partial_{p}H\right)}^{2}+{\left(\partial_{n}H\right)}^{2}}}{{\left(\partial_{q}H\right)}^{2}+{\left(\partial_{p}H\right)}^{2}+{\left(\partial_{n}H\right)}^{2}},$$
$$G_{E}^{nn}=\frac{b\left[{\left(\partial_{q}H\right)}^{2}+{\left(\partial_{p}H\right)}^{2}\right]}{{\left(\partial_{q}H\right)}^{2}+{\left(\partial_{p}H\right)}^{2}+{\left(\partial_{n}H\right)}^{2}},$$
with
(23)$$\left\{\begin{array}{c} b=\frac{\psi_{n}}{\chi S^{\prime }\left(n\right)}\frac{{\left(\partial_{q}H\right)}^{2}+{\left(\partial_{p}H\right)}^{2}+{\left(\partial_{n}H\right)}^{2}}{{\left(\partial_{q}H\right)}^{2}+{\left(\partial_{p}H\right)}^{2}},\\ c=\frac{\psi_{p}\partial_{q}H-\psi_{q}\partial_{p}H}{\chi S^{\prime }\left(n\right)}\frac{\sqrt{{\left(\partial_{q}H\right)}^{2}+{\left(\partial_{p}H\right)}^{2}+{\left(\partial_{n}H\right)}^{2}}}{{\left(\partial_{q}H\right)}^{2}+{\left(\partial_{p}H\right)}^{2}}. \end{array}\right.$$

In the system at hand, due to the nature of the dissipation terms $\psi_{q}$ and $\psi_{p}$ in (Equation 8) and (Equation 15), it is easy to see that $\psi_{q}\partial_{p}H-\psi_{p}\partial_{q}H=0$, so that the term *c* in (Equation 23) simply vanishes. This is definitely not a general condition, rather it requires the two variables *q* and *p* to be dissipated formally in the same way, and the Hamiltonian *H* to be symmetric under the exchange of *q* and *p*. A simple counterexample for which c≠0 is illustrated in the Appendix, with the result (Equation 52).

The set of components of the metric tensor $G_{E}$ may be finally expressed in a much more compact, and yet explicit form: putting together the expression $H\left(q,p,n\right)$, the values of *b* in (Equation 23), and c=0, after some algebra, the expressions
(24)$$\left\{\begin{array}{cc} \; & G_{E}^{qq}=\frac{8\alpha^{2}}{\chi kS^{\prime }\left(n\right)}\frac{q^{2}n^{2}}{{\left(q^{2}+p^{2}\right)}^{2}},\;\;G_{E}^{qp}=\frac{8\alpha^{2}}{\chi kS^{\prime }\left(n\right)}\frac{qpn^{2}}{{\left(q^{2}+p^{2}\right)}^{2}},\;\;G_{E}^{pp}=\frac{8\alpha^{2}}{\chi kS^{\prime }\left(n\right)}\frac{p^{2}n^{2}}{{\left(q^{2}+p^{2}\right)}^{2}},\\ \; & G_{E}^{qn}=-\alpha \frac{nq}{\chi S^{\prime }\left(n\right)},\;\;G_{E}^{pn}=-\alpha \frac{np}{\chi S^{\prime }\left(n\right)},\;\;G_{E}^{nn}=\frac{k}{8\chi S^{\prime }\left(n\right)}{\left(q^{2}+p^{2}\right)}^{2} \end{array}\right.$$
are written.

With reference to the Appendix, one attains a third equation from Equation (Equation 50), namely,
(25)$$\psi_{q}\partial_{q}H+\psi_{p}\partial_{p}H+\psi_{n}\partial_{n}H=0.$$ This last condition expresses the conservation of the total Hamiltonian *H* when the relationships (Equation 50) are enforced, which is precisely what is required by theory (namely, dissipation does not alter the whole amount of energy).

### 4.2. The Total Hamiltonian

Considering Equation (Equation 8), we fix
(26)$$\left\{\begin{array}{c} \psi_{q}=-\alpha nq,\\ \psi_{p}=-\alpha np,\\ \psi_{n}=\frac{k}{8}{\left(q^{2}+p^{2}\right)}^{2} \end{array}\right.$$
therefore, $\partial_{q}H=\frac{1}{2}q\left(q^{2}+p^{2}\right)$ and $\partial_{p}H=\frac{1}{2}p\left(q^{2}+p^{2}\right)$, which imply that the total Hamiltonian reads:(27)$$H\left(q,p,n\right)=H_{0}\left(q^{2}+p^{2}\right)+U\left(n\right),$$
with $H_{0}\left(q^{2}+p^{2}\right)$ being the free radiation Hamiltonian defined in Equation (Equation 12). In order to determine U(n), we need to take into account Equation (Equation 25):(28)$$-\alpha nq\frac{1}{2}q\left(q^{2}+p^{2}\right)-\alpha np\frac{1}{2}p\left(q^{2}+p^{2}\right)+\frac{k}{8}{\left(q^{2}+p^{2}\right)}^{2}U^{\prime }\left(n\right)=0,$$
whence
(29)$$U\left(n\right)=\frac{2\alpha }{k}n^{2}+U_{0}.$$

### 4.3. Entropy and Equilibrium States

We are now in the position to explicitly write the free energy *F*
$$F\left(q^{2}+p^{2},n\right)=H_{0}\left(q^{2}+p^{2}\right)+U\left(n\right)+\chi S\left(n\right):$$
as the expression (Equation 29) is used, one has
(30)$$F\left(q^{2}+p^{2},n\right)=\frac{1}{8}{\left(q^{2}+p^{2}\right)}^{2}+\frac{2\alpha }{k}n^{2}+U_{0}+\chi S\left(n\right).$$

Equilibrium states of the system must satisfy the condition
(31)$$\delta F=\partial_{q}F\delta q+\partial_{p}F\delta p+\partial_{n}F\delta n=0,$$
that is:(32)$$q_{eq}=p_{eq}=0,\;\;S^{\prime }{\left(n\right)|}_{n_{eq}}=-\chi \frac{4\alpha }{k}n_{eq}.$$

As expected, different forms of entropy function correspond to different equilibrium points.

Some lines ago we anticipated that χ→0 suppresses the metric part of the dynamics. This means putting oneself in the condition of an equilibrium reached without populating the atomic excited level (e.g., at $0\;{}^{\circ }$ K temperature), which does not mean turning off the matter–radiation coupling.

Before going to the conclusions, it is important to note that α and *k* appear everywhere as a ratio. It would make sense to introduce an *always finite* constant κ so that k=κα. This would reduce the non-dissipative condition (Equation 10) to the much simpler α→0, that is, again, a statement about interactions, while χ→0 would be a statement about the equilibrium around which we are working.

## 5. The CMS in Madelung Variables

In this Section, we are going to present the same CMS as before, but describing the electromagnetic phasor ψ via its square-root amplitude and phase variables ρ and ϕ as defined in (Equation 9), instead of the two quantities *q* and *p*. The variables $\left(\phi ,\rho \right)$ are related to the $\left(q,p\right)$ ones via the transformation:(33)$$\left\{\begin{array}{cc} \; & q=\sqrt{2\rho }\cos\phi ,\\ \; & p=-\sqrt{2\rho }\sin\phi , \end{array}\right.\;\;\left\{\begin{array}{cc} \; & \phi =-\arctan\left(\frac{p}{q}\right),\\ \; & \rho =\frac{1}{2}\left(q^{2}+p^{2}\right). \end{array}\right.$$ These are invertible and smooth, except for ρ=0, that is, the singular point at which no radiation exists at all.

It is easy to see that the new set of variables $\left(\phi ,\rho ,n\right)$ evolves according to the ODEs
(34)$$\left\{\begin{array}{cc} \; & \dot{\phi }=\rho ,\\ \; & \dot{\rho }=-2\alpha n\rho ,\\ \; & \dot{n}=\frac{k}{2}\rho^{2}, \end{array}\right.$$
as the system $\left(q,p,n\right)$ undergoes (Equation 8). The α→0 and k→0 limit of (Equation 34) gives the non-dissipative approximation of those equations:(35)$$\left\{\begin{array}{cc} \; & \dot{\phi }=\rho ,\;\;\dot{\rho }=0,\\ \; & \dot{n}=0. \end{array}\right.$$

Equation (Equation 35) suggest the angle–frequency nature of the variables ϕ and ρ in the Hamiltonian limit. According to them, ρ is a constant of motion, while $\phi \left(t\right)=\rho t+\phi \left(0\right)$ evolves linearly with time. Dissipation, as in (Equation 34), slows down the run of $\phi \left(t\right)$ as it consumes ρ in favour of the population *n*. Equations (Equation 35) form a Hamiltonian system, provided one uses the symplectic tensor
(36)$$J^{\prime }\left(\phi ,\rho ,n\right)=\left(\begin{array}{ccc} 0 & 1 & 0\\ -1 & 0 & 0\\ 0 & 0 & 0 \end{array}\right)$$
and the Hamiltonian
(37)$$H_{0}\left(\rho \right)=\frac{1}{2}\rho^{2}.$$ This $H_{0}\left(\rho \right)$ is not conserved under the motion (Equation 34): the extension
(38)$$H\left(\phi ,\rho ,n\right)=\frac{1}{2}\rho^{2}+\frac{2\alpha }{k}n^{2}+U_{0}$$
of (Equation 37) closes the system energetically, because $H\left(\phi ,\rho ,n\right)$ is conserved under the motion (Equation 34) (moreover, this *H* does not truely depend on ϕ). Nonetheless, it is not possible to generate the whole dynamics (Equation 34) via just the Hamiltonian (Equation 38) and the PB given by (Equation 36): the system (Equation 34) can be rather represented as a *complete metriplectic system* by adding to $\left\{\cdot ,H\left(\phi ,\rho ,n\right)\right\}$ a metric bracket expressed in terms of ϕ, ρ and *n*.

In order to complete the system (Equation 34) as a CMS, one may either re-formulate ex novo the whole problem, as achieved in the $\left(q,p,n\right)$ variables, repeating everything in the new $\left(\phi ,\rho ,n\right)$, through the Equation (Equation 34), the Hamiltonian (Equation 38) and the PB given by the Jacobi tensor (Equation 36); or, as will be the case here, by appreciating the *tensor nature* of the metriplectic laws [6]
(39)$${\dot{x}}^{\mu }=J^{\mu \nu }\left(x\right)\partial_{\nu }H\left(x\right)+\chi G^{\mu \nu }\left(x\right)\partial_{\nu }S\left(x\right)$$
under any diffeomorphic change of variables $x\mapsto y\left(x\right)$. Indeed, provided *H* and *S* are *scalar quantities* under the transformations $x\mapsto y\left(x\right)$, one may perform the variable change, and obtain:(40)$${\dot{y}}^{\rho }=\frac{\partial y^{\rho }}{\partial x^{\mu }}\frac{\partial y^{\sigma }}{\partial x^{\nu }}J^{\mu \nu }\left(x\left(y\right)\right)\frac{\partial H\left(y\right)}{\partial y^{\sigma }}+\chi \frac{\partial y^{\rho }}{\partial x^{\mu }}\frac{\partial y^{\sigma }}{\partial x^{\nu }}G^{\mu \nu }\left(x\right)\frac{\partial S\left(y\right)}{\partial y^{\sigma }}.$$ This (Equation 40) is equivalent to the initial system (Equation 39), and it may be put into an explicitly CMS-like form
(41)$${\dot{y}}^{\rho }=J^{{}^{\prime }\rho \sigma }\left(y\right)\frac{\partial H\left(y\right)}{\partial y^{\sigma }}+\chi G^{\prime \rho \sigma }\left(x\right)\frac{\partial S\left(y\right)}{\partial y^{\sigma }},$$
provided the identifications
(42)$$\left\{\begin{array}{cc} \; & J^{{}^{\prime }\rho \sigma }\left(y\right)=\frac{\partial y^{\rho }}{\partial x^{\mu }}\left(x\left(y\right)\right)\frac{\partial y^{\sigma }}{\partial x^{\nu }}\left(x\left(y\right)\right)J^{\mu \nu }\left(x\left(y\right)\right),\\ \; & G^{{}^{\prime }\rho \sigma }\left(y\right)=\frac{\partial y^{\rho }}{\partial x^{\mu }}\left(x\left(y\right)\right)\frac{\partial y^{\sigma }}{\partial x^{\nu }}\left(x\left(y\right)\right)G^{\mu \nu }\left(x\left(y\right)\right) \end{array}\right.$$
are achieved. Equation (Equation 42) state precisely that the Jacobi matrix *J* and the semimetric matrix *G* transform into the matrices $J^{\prime }$ and $G^{\prime }$, respectively, *as rank-2 tensors* under any change of variables $x\mapsto y\left(x\right)$.

As the change of variables $\left(q,p,n\right)\mapsto \left(\phi ,\rho ,n\right)$ in (Equation 33) is applied to the calculation of $J^{{}^{\prime }\rho \sigma }=\frac{\partial y^{\rho }}{\partial x^{\mu }}\frac{\partial y^{\sigma }}{\partial x^{\nu }}J^{\mu \nu }$, the matrix *J* in (Equation 16) is transformed into the matrix $J^{\prime }$ in (Equation 36), meaning that (Equation 33) is a *canonical change of variables*.

In order to find the semimetric matrix $G^{\prime }$ completing the CMS that represents the system (Equation 34), one applies the law $G^{{}^{\prime }\rho \sigma }=\frac{\partial y^{\rho }}{\partial x^{\mu }}\frac{\partial y^{\sigma }}{\partial x^{\nu }}G^{\mu \nu }$ to the matrix $G\overset{\mathrm{def}}{=}G_{E}\left(q,p,n\right)$ in (Equation 24) with the gradients of (Equation 33), obtaining:$$\left\{\begin{array}{cc} & G^{\prime \phi \phi }=2\partial_{q}\phi \partial_{p}\phi G^{qp}+{\left(\partial_{q}\phi \right)}^{2}G^{qq}+{\left(\partial_{p}\phi \right)}^{2}G^{pp},\\ & G^{\prime \phi \rho }=\partial_{q}\phi \partial_{q}\rho G^{qq}+\partial_{p}\phi \partial_{p}\rho G^{pp}+\left(\partial_{q}\phi \partial_{p}\rho +\partial_{p}\phi \partial_{q}\rho \right)G^{qp},\\ & G^{\prime \phi n}=\partial_{q}\phi \partial_{n}nG^{qn}+\partial_{p}\phi \partial_{n}nG^{pn},\\ & G^{\prime \rho \rho }={\left(\partial_{q}\rho \right)}^{2}G^{qq}+{\left(\partial_{p}\rho \right)}^{2}G^{pp}+2\partial_{q}\rho \partial_{p}\rho G^{qp},\\ & G^{\prime \rho n}=\partial_{q}\rho \partial_{n}nG^{qn}+\partial_{p}\rho \partial_{n}nG^{pn},\\ & G^{\prime nn}={\left(\partial_{n}n\right)}^{2}G^{nn}, \end{array}\right.$$
that is
(43)$$\left\{\begin{array}{cc} \; & G^{\prime \phi \phi }=0,\;\;G^{\prime \phi \rho }=0,\;\;G^{\prime \phi n}=0,\\ \; & G^{\prime \rho \rho }=\frac{8\alpha^{2}n^{2}}{\chi kS^{\prime }\left(n\right)},\\ \; & G^{\prime \rho n}=-\frac{2\alpha n\rho }{\chi S^{\prime }\left(n\right)},\\ \; & G^{\prime nn}=\frac{k\rho^{2}}{2\chi S^{\prime }\left(n\right)}. \end{array}\right.$$

This is the matrix to be used in the composition of the symmetric bracket $\chi \left(\cdot ,S\left(n\right)\right)$ to obtain the dissipative terms
(44)$$\eta_{\phi }=0,\;\;\eta_{\rho }=-2\alpha n\rho ,\;\;\eta_{n}=\frac{k}{2}\rho^{2}$$
to be added to the nondissipative ODEs (Equation 35) in order to obtain the dissipative ones (Equation 34). In particular, it is straightforward to check:(45)$$\eta_{\phi }=\chi G^{\prime \phi n}S^{\prime }\left(n\right),\;\;\eta_{\rho }=\chi G^{\prime \rho n}S^{\prime }\left(n\right),\;\;\eta_{n}=\chi G^{\prime nn}S^{\prime }\left(n\right),$$
expressions to which $\chi \left(\phi ,S\left(n\right)\right)$, $\chi \left(\rho ,S\left(n\right)\right)$ and $\chi \left(n,S\left(n\right)\right)$ reduce, respectively. Last but not least, one observes that the Hamiltonian compatibility condition $G^{\prime }\cdot \nabla^{\prime }H\left(\phi ,\rho ,n\right)=0$ is satisfied, considering the expression (Equation 38) for the Hamiltonian, as $G^{\prime }$ in (Equation 43) is used, and $\nabla^{\prime }$ is the gradient with respect to ϕ, ρ and *n*.

Wrapping up what has been found in the present Section, we may state that the TPA system can also be regarded as a CMS when the radiation is represented via the phase–amplitude variables ϕ and ρ, defined in (Equation 9), and that the new expression of the semimetric matrix $G^{\prime }\left(\phi ,\rho ,n\right)$ is obtained by trasforming the matrix $G_{E}\left(q,p,n\right)$ as a rank-2 tensor under the transformations (Equation 33).

## 6. Conclusions

This work applies metriplectic theory and the technique of Leibniz algebrae to a dissipative nonlinear optical phenomenon: the two-photon absorption by a two-level atom with negligible spontaneous emission. Once the physical problem was formulated in terms of the conservative part $H_{0}$ of the total Hamiltonian *H*, the metric tensor *G* and the metriplectic brackets <<·,·>>, we found the mathematical expression of *H* as function of the dynamical variables *q*, *p* and *n*, representing the electric phasor ψ via its real components. In particular, we have found the internal part *U* of *H*, which depends only on the second-level population *n*. We have also found the free energy *F* and the equilibrium states, varying with the definition of entropy, as expected.

Use was then made of the rank-2 tensor nature of the Jacobi matrix *J* and of the semimetric matrix *G* to offer another representation of the CMS, in which the electric phasor ψ is assigned by the real couple $\left(\phi ,\rho \right)$ of its phase ϕ and square-rooted amplitude ρ.

We believe that this manuscript opens the way to an ambitious research program in which the metriplectic formalism is used to explore irreversibility in nonlinear optics. Applications may be envisaged in regimes including an interplay between interaction and absorptions, including multi-modal regimes in lasers and fibers and systems with topologically non-trivial features, arising from engineered distributions of gain and loss, such as those supporting P-T symmetry or disorder. In all these cases, finding Casimir as generators of non-Hamiltonian motions may unveil novel regimes and phase transitions which are not tackled by conventional methods.

## Figures and Tables

**Figure 1 entropy-24-00506-f001:**
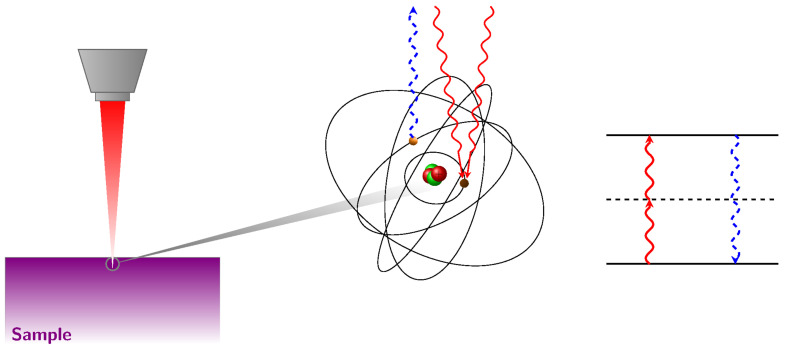
Pictorial sketch of absorbtion of two photons in a two-level atom. In our system, Equation (Equation 6) does not have terms of spontaneous emission, considered negligible. This is here represented by the dashed blue line, not present in our model.

## Data Availability

Not applicable.

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
