# Peer review of "Metriplectic Structure of a Radiation–Matter-Interaction Toy Model"

_entropy, 2022, doi:10.3390/e24040506_

Round 1
Reviewer 1 Report
The manuscript describes a non-standard method for describing dissipative processes.
This method is based on the so-called Metriplectic Structure.
Some general comments about the Metriplectic Structure approach:
Thise approach allows us to obtain some new results by introducing some generalization of the concept of a Hamiltonian or gradient system.
In my opinion, this approach has a significant limitation, which is due to the fact that a wide class of dissipative systems cannot be considered within the framework of this approach.
In addition, the metriplectic Leibniz brackets (19), in standard algebraic approach to classical mechanics and quantum mechanics correspond to the sum of two
products (Lie and Jordan products), which are considered independently.
About the manuscript:
The manuscript is well written and contains interesting results.
However, it can be improved for the convenience of readers.
In equations and formulas, dependencies and conditions should be written more accurately.
For example, equation (3): on the left in the equality, the limits of functions are considered with respect to the alpha parameter.
However, in the functions themselves, the dependence on this parameter is not indicated.
The manuscript should be published.
Taking into account the proposed comments will make it easier for readers of the article to understand the work.
Author Response
It is our pleasure to response point-by-point (R) to Referee's comments (C), that are in quotes.
In the new version of the manuscript uploaded, all the changes as in bold.
1C. "Some general comments about the Metriplectic Structure approach: This approach allows us to obtain some new results by introducing some generalization of the concept of a Hamiltonian or gradient system. In my opinion, this approach has a significant limitation, which is due to the fact that a wide class of dissipative systems cannot be considered within the framework of this approach. In addition, the metriplectic Leibniz brackets (19), in standard algebraic approach to classical mechanics and quantum mechanics correspond to the sum of two products (Lie and Jordan products), which are considered independently."
1R. This comments are useful and interesting. Our opinion is that, even if a wide class of dissipative systems cannot be cast into metriplectic formalism, yet we have shown some important examples of systems than can. Most of all, the physical system at hand, that of the dissipative representation of two-photon absorption, is among those systems that allow a metriplectic representation. An observation about this point is now reported at page 2, first column, of the re-submitted text. The fact that the metriplectic bracket is the sum of a Lie product and a Jordan product is true: not only this catches perfectly the union of a Hamiltonian and a metric dynamics, it also indicates the abstract nature of this formalism, possibly equally suitable for classical and quantum systems.
2C. "The manuscript is well written and contains interesting results.
However, it can be improved for the convenience of readers.
In equations and formulas, dependencies and conditions should be written more accurately.
For example, equation (3): on the left in the equality, the limits of functions are considered with respect to the alpha parameter. However, in the functions themselves, the dependence on this parameter is not indicated."
2R. Thanks for the very useful criticism! Indeed, we have tried to preface a physical reasoning to the mathematical formulation, so that the reader may better appreciate what we try to do. In particular, in the introduction we have inserted long arguments about why one should conceive the process of electromagnetic energy absorption by some medium as a dissipative process, and hence treat it with metriplectic tools. Clarifications and corrections have been inserted in Section II (see the new version of Equation (3) in particular).

Reviewer 2 Report
Authors are trying to develop a new formalism of metriplectic structure and apply it to study the model of the radiation-matter interaction. The method extends the Hamiltonian formalism of the system evolution, and the authors construct the toy model of nonlinear optical phenomena like the photon interaction with two-level system. The paper can be published but it is worthy to add some discussion related with experiments in quantum optics to make more clear advantages of the suggested formalism.
Author Response
It is our pleasure to response point-by-point (R) to Referee's comments (C), that are in quotes.
In the new version of the manuscript uploaded, all the changes as in bold.
1C. “Authors are trying to develop a new formalism of metriplectic structure and apply it to study the model of the radiation-matter interaction. The method extends the Hamiltonian formalism of the system evolution, and the authors construct the toy model of nonlinear optical phenomena like the photon interaction with two-level system. The paper can be published but it is worthy to add some discussion related with experiments in quantum optics to make more clear advantages of the suggested formalism. “
1R. This discussion, in a short form, has been added at the end of Section VI, Conclusions, page 10, second column.
